environmental chemistry/nanotechnology/
materials science

dye wastewater, photocatalytic degradation,
methylene blue dye, sol–gel method,
iron titanate

**Author for correspondence:**
Z. Z. Vasiljevic
e-mail: zorka.djuric@itn.sanu.ac.rs

This article has been edited by the Royal Society of Chemistry, including the commissioning, peer review process and editorial aspects up to the point of acceptance.

# Photocatalytic degradation of methylene blue under natural sunlight using iron titanate nanoparticles prepared by a modified sol–gel method

Z. Z. Vasiljevic[1], M. P. Dojcinovic[2], J. D. Vujancevic[1],
I. Jankovic-Castvan[3], M. Ognjanovic[4], N. B. Tadic[5],
S. Stojadinovic[5], G. O. Brankovic[2] and M. V. Nikolic[2]

[1]Institute of Technical Sciences of SASA, Belgrade, Serbia
[2]Institute for Multidisciplinary Research, [3]Faculty of Technology and Metallurgy, [4]Institute of Nuclear Sciences Vinca, and [5]Faculty of Physics, University of Belgrade, Belgrade, Serbia

ZZV, 0000-0002-7817-8648

The aim of this work was to synthesize semiconducting oxide nanoparticles using a simple method with low production cost to be applied in natural sunlight for photocatalytic degradation of pollutants in waste water. Iron titanate ($Fe_2TiO_5$) nanoparticles with an orthorhombic structure were successfully synthesized using a modified sol–gel method and calcination at 750°C. The as-prepared $Fe_2TiO_5$ nanoparticles exhibited a moderate specific surface area. The mesoporous $Fe_2TiO_5$ nanoparticles possessed strong absorption in the visible-light region and the band gap was estimated to be around 2.16 eV. The photocatalytic activity was evaluated by the degradation of methylene blue under natural sunlight. The effect of parameters such as the amount of catalyst, initial concentration of the dye and pH of the dye solution on the removal efficiency of methylene blue was investigated. $Fe_2TiO_5$ showed high degradation efficiency in a strong alkaline medium that can be the result of the facilitated formation of OH radicals due to an increased concentration of hydroxyl ions.

# 1. Introduction

The major worldwide challenge for the twenty-first century is to supply and ensure safe water for the entire ecosystem. Rapid industrialization growth is the major cause of water pollution. Dyes are a class of organic compounds, widely used in textiles, printing and food industry. Dye effluents have a considerable negative influence on the environment, and most of them are highly toxic and non-biodegradable [1,2]. Methylene blue (MB) is a phenothiazine derivative, used for dying textiles, and it is highly toxic and carcinogenic [3]. Conventional methods like adsorption, ozonation, etc., have been used for removing these highly toxic compounds; however, due to different limitations, these pollutants cannot be eliminated from waste water [4]. Advanced oxidation processes (AOPs) are a class of oxidation techniques in which organic contaminants are degraded to harmless products. Namely, in this process, ·OH radicals produced *in situ*, initiate oxidation reactions ending with complete mineralization of $CO_2$ and $H_2O$ [5,6]. In this regard, semiconductor photocatalysis is a promising technique for the degradation of organic pollutants. There are several reasons why using semiconductors as activators in AOPs has an advantage: (i) they are low-cost and non-toxic; (ii) their electronic and optical properties can be modified by size reduction, doping or sensitizers; (iii) the multi-electron transfer process is facilitated; and (iv) they are recyclable without substantial loss in photocatalytic activity [5,7]. $TiO_2$ has been extensively investigated as a photocatalyst for the decomposition of organic pollutants [8,9]. Despite its good chemical stability and low production cost, one major difficulty using this photocatalyst is its large band gap limiting photocatalytic activity in the visible-light region. Thus, it is necessary to develop materials which will absorb in the visible-light regime and prevent electron–hole recombination. Haematite ($Fe_2O_3$) with a band gap of 2.1–2.2 eV, is also a promising material for use in photocatalysis due to its ability to absorb light up to 600 nm, high stability in an oxidative environment, abundance and low-cost fabrication [10]. However, due to low conductivity and short hole diffusion length, its photocatalytic performance is still limited [11,12]. Iron titanate ($Fe_2TiO_5$), a hybrid of $TiO_2$ and $Fe_2O_3$, is a narrow band gap ($E_g$ approx. 2.2 eV) semiconductor absorbing in the visible-light spectrum, similar to $Fe_2O_3$. It is an inexpensive, non-toxic material with high chemical stability. Additionally, its electronic and atomic structures are similar to $TiO_2$ [13,14], thus recombination of photo-generated electrons and holes could be suppressed facilitating transport to hydroxyl groups on the particle surface [15].

In this study, $Fe_2TiO_5$ nanoparticles were synthesized by a modified sol–gel method and we have performed a detailed characterization of their structure, morphology, texture and chemistry, optical and electronic properties. The photocatalytic activity of prepared $Fe_2TiO_5$ nanoparticles was evaluated by the degradation of a cationic dye—methylene blue (MB) exposed to 4 h of natural sunlight. Several operating parameters, such as dose of photocatalyst and the initial pH of the solutions, were also evaluated.

# 2. Materials and methods

## 2.1. Reagents and materials

Methylene blue, iron(III) nonahydrate ($Fe(NO_3)_3 \cdot 9H_2O$ ACS reagent, purity greater than or equal to 98%), titanium isopropoxide—($Ti(OCH(CH_3)_2)$ purity 98%) oxalic acid (Puriss, purity greater than or equal to 99%) and cetyltrimethylammonium bromide (CTAB, purity greater than or equal to 98%) were purchased from Sigma Aldrich (Merck, Darmstadt, Germany). All chemicals were used as received, without further purification.

## 2.2. Synthesis of iron titanate nanoparticles

$Fe_2TiO_5$ nanoparticles were synthesized by a modified sol–gel method using $Fe(NO_3)_3 \cdot 9H_2O$ and $Ti(OCH(CH_3)_2)$ as starting reagents, oxalic acid as a chelating agent and CTAB as a surfactant. Details of the synthesis procedure are described in [16]. The obtained powder was calcined at 750°C for 3 h to obtain the desired $Fe_2TiO_5$ nanoparticles.

## 2.3. Characterization

The prepared sample was characterized by various techniques. The morphology was examined by transmission electron microscopy (TEM) and field emission electron microscopy (FESEM) on JEM-2100

200 kV (Japan) and TESCAN MIRA3 XM (Czech Republic) devices, respectively. The crystal structure and phase identification were investigated by X-ray diffraction (XRD) analysis using Ni-filtered Cu K$\alpha$ radiation with the wavelength of 1.54178 Å, in the range of $2\theta = 10 - 90°$, with the step of 0.05 s and acquisition rate of 1 ° min$^{-1}$ on a Rigaku Ultima IV diffractometer (Japan). Fourier transform infrared (FT-IR) spectra were recorded on an FT-IR Nicolet 6700 ATR device (Thermo Fisher, UK) in the range 400– 4000 cm$^{-1}$. UV-vis diffuse reflectance spectra (DRS) were measured on a Shimadzu UV-2600 device with an ISR2600 Plus Integrating sphere attachment (Japan) in the measuring range 200–1400 nm.

Photoluminescence spectra at room temperature were measured on a Horiba Jobin Yvon Fluorolog FL3–22 spectrofluorometer (Japan), using a xenon 450 W lamp as the light source. It was coupled to a double grating monochromator in a wavelength range 220–600 nm. Emission spectra were measured using a double grating monochromator in the 350–650 nm range and side-on photomultiplier (Hamamatsu, model 928P, Japan). The measured spectra were corrected for the spectral response of the measuring system and spectral distribution of the xenon lamp.

Zeta potential and particle size by dynamic light scattering (DLS) at $25 \pm 0.1$°C were determined for a 0.1 mg ml$^{-1}$ suspension of Fe$_2$TiO$_5$ powder dispersed ultrasonically in deionized water and measured in a disposable zeta cell (DTS 1070) of a NanoZS90 (Malvern, UK) device with a 4 mW He-Ne laser source ($\lambda = 633$ nm). The pH was varied between 3 and 12 by adding 0.1 M aqueous solutions of HCl and NaOH. Samples were measured after 1 h of equilibrium time at a constant ionic strength of 0.01 M set by NaCl. All measurements were performed at a given kinetic state followed by a minute of relaxation. The average value of the hydrodynamic diameter was calculated as the number weighted size from fits of the correlation functions.

The specific surface area (SSA) and the porous properties of the photocatalyst were determined based on N$_2$ adsorption–desorption isotherms measured using a Micromeritics ASAP 2020 instrument (USA). Samples were degassed at 150°C for 10 h under reduced pressure. SSA of samples was calculated according to the Brunauer–Emmett–Teller (BET) method from the linear part of the nitrogen adsorption isotherm. The total pore volume ($V_{tot}$) was given at $p/p_0 = 0.998$. The volume of the mesopores was calculated according to the Barrett–Joyner–Halenda method from the desorption branch of the isotherm. The volume of micropores was calculated from the alpha-S plot.

## 2.4. Photocatalytic experiments

The photocatalytic activity of Fe$_2$TiO$_5$ nanoparticles was evaluated by the photodegradation of methylene blue (MB) aqueous solution under direct solar radiation in July 2019 between 10.00 and 14.00, where the average daily temperature was $31 \pm 2$°C. Before irradiation, the solutions were magnetically stirred in dark for 60 min to achieve adsorption–desorption equilibrium between photocatalyst and pollutant. A stock solution of 1000 mg l$^{-1}$ MB was prepared by dissolving MB in 1 l of distilled water. Different amounts of Fe$_2$TiO$_5$ catalyst (10, 30 and 50 mg l$^{-1}$) were tested for the degradation of MB solution. Samples were collected in appropriate time intervals (up to 4 h) and were centrifuged to remove the catalyst. The removal of MB was determined based on the absorption at 662 nm by using a UV–vis spectrophotometer. Then, the absorption was converted to the concentration through the standard calibration curve.

# 3. Results and discussion

## 3.1. Iron titanate structure

Figure 1*a* shows the measured XRD pattern of the obtained Fe$_2$TiO$_5$ powder. Structural parameters were refined using the Rietveld method and the GSAS II software package [18]. Pseudobrookite was refined assuming the *Cmcm* ($D_{2h}^{17}$) space group starting with a random distribution of TiO$_6$ and FeO$_6$ octahedra, where 67% iron ions are both in 4c and 8f sites of the FeO$_6$ octahedra [19]. As starting unit cell parameters, we used the values previously determined for Fe$_2$TiO$_5$ nanocrystalline powder obtained by solid-state synthesis [20]. All diffraction peaks can be indexed to the orthorhombic phase of Fe$_2$TiO$_5$ with space group $D_{2h}^{17}$ (*Cmcm*) without any peaks originating from impurities. The Williamson–Hall method [21] was applied to estimate the crystallite size of Fe$_2$TiO$_5$ powder as 48.77 nm with a positive tensile strain of 0.00108. The determined unit cell parameters and atomic positions are given in table 1. Refinement of ion occupancy showed that differing from the previously obtained fully disordered structure [19] iron ions showed a slight preference for 4c sites occupying

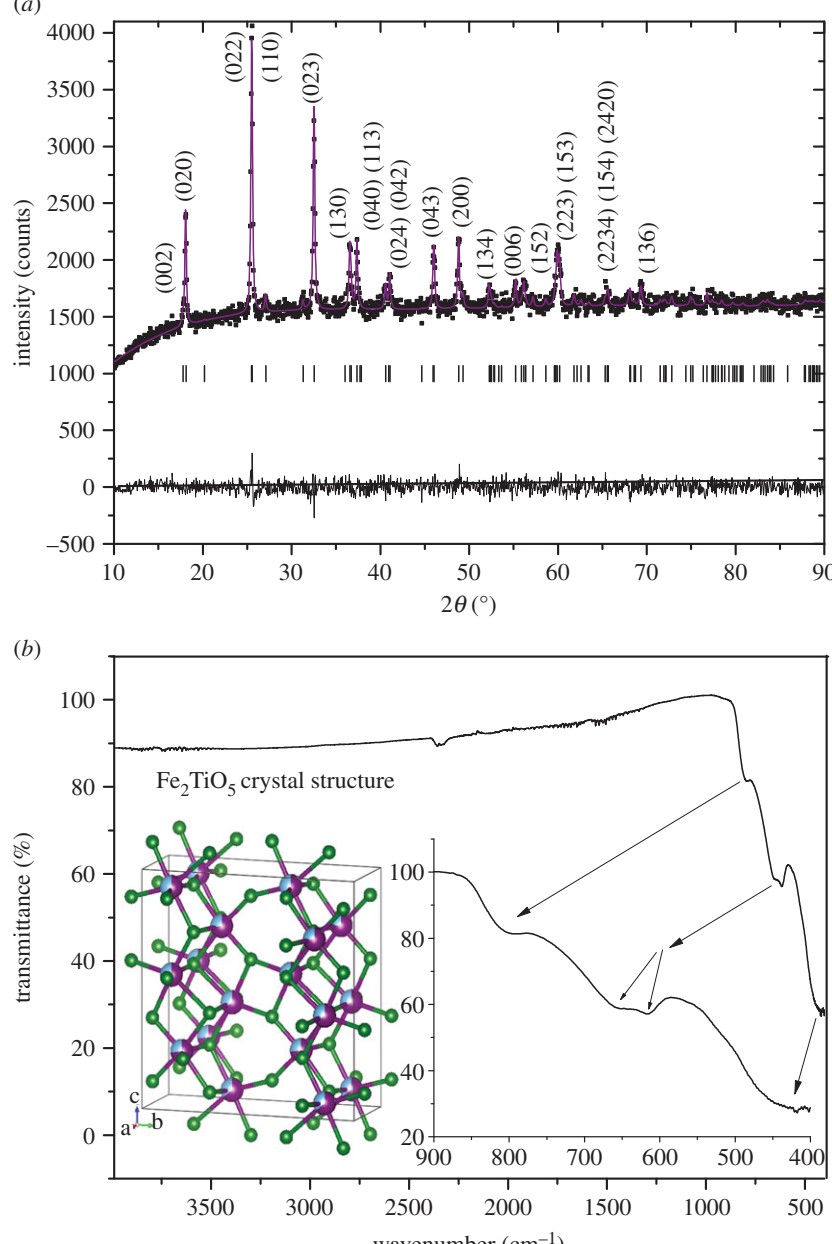

**Figure 1.** X-ray diffraction pattern of synthesized Fe$_2$TiO$_5$ nanocrystalline powder and Rietveld analysis (*a*) and FT-IR transmittance spectrum of Fe$_2$TiO$_5$ nanocrystalline powder; inset: crystal structure was drawn using VESTA [17] (*b*).

72%, while 64% iron ions are on 8f sites. Such an iron preference was previously noted by Guo *et al.* [22] of 72% and Rodrigues *et al.* [23] of 75%. The crystal structure drawn using these parameters and the VESTA software package [17] is shown as an inset in figure 1*b* composed of randomly distributed TiO$_6$ and FeO$_6$ octahedra. Ti ions are blue (one third), iron ions are purple (two thirds) parts of the balls in the polyhedra representing metal ions, while green are oxygen ions.

Structural properties such as the vibrational frequency of the bonds in Fe$_2$TiO$_5$ were further characterized by FT-IR spectroscopy (figure 1*b*). In our measured spectra, we can note only peaks originating from Fe$_2$TiO$_5$, at approximately 425, 615, 655 and 800 cm$^{-1}$ representing vibrations of Fe–O and Ti–O bonds [24–27].

## 3.2. Morphology

The Fe$_2$TiO$_5$ powder particle size and morphology was examined by FESEM and TEM. The FESEM image shown in figure 2*a* reveals that Fe$_2$TiO$_5$ powder consisted of relatively uniformly sized

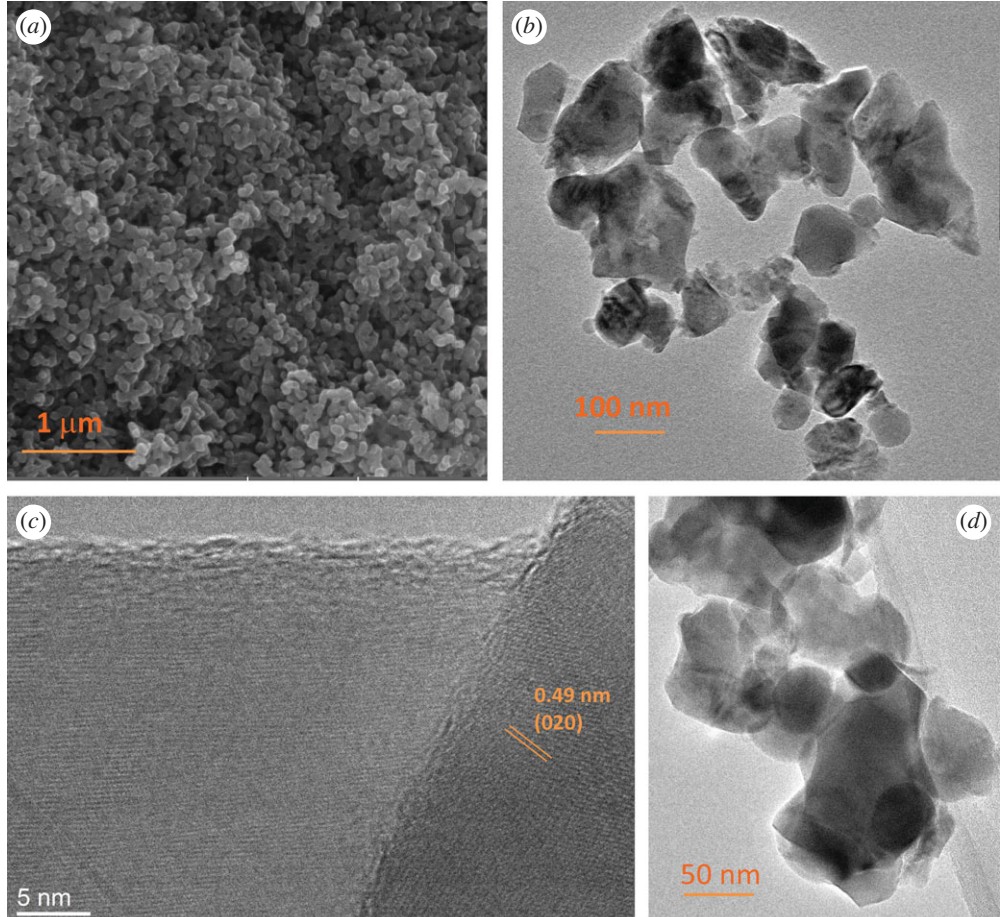

**Figure 2.** FESEM, magnification 50 000 (*a*), TEM images (*b*) magnification 25 000, (*d*) magnification 50 000 and HRTEM and FFT images (*c*) magnification 500 000 of $Fe_2TiO_5$ powder.

**Table 1.** Unit cell parameters and atomic positions determined for $Fe_2TiO_5$, wR = 2.987%.

| pseudobrookite ($Fe_2TiO_5$) space group *Cmcm, a* = 3.73055(28), *b* = 9.7927(12), *c* = 9.9766(10) Å; crystallite size 48.77 nm, microstrain 1937.2, (wR = 2.797%) | | | | | | |
|---|---|---|---|---|---|---|
| atom | site | *x* | *y* | *z* | occupancy | $U_{iso}$ |
| $Fe_1$ | 8f | 0.00000 | 0.1388(11) | 0.5667(13) | 0.64 | 0.012(5) |
| $Ti_1$ | 8f | 0.00000 | 0.1388 | 0.5667 | 0.36 | 0.012 |
| $Fe_2$ | 4c | 0.00000 | 0.1875(13) | 0.25000 | 0.72 | 0.008(8) |
| $Ti_2$ | 4c | 0.00000 | 0.1875 | 0.25000 | 0.28 | 0.008 |
| $O_1$ | 4c | 0.00000 | 0.757(5) | 0.25000 | 1.000 | 0.011(18) |
| $O_2$ | 8f | 0.00000 | 0.050(3) | 0.117(3) | 1.000 | 0.015(13) |
| $O_3$ | 8f | 0.00000 | 0.319(4) | 0.084(4) | 1.000 | 0.046(12) |

nanoparticles with a rhombic shape ranging from 28 to 83 nm and average diameter determined from 72 particles in TEM images of 49.7 nm ± 12.6 (standard deviation). This morphology was confirmed by TEM (figure 2*b,d*), showing that the $Fe_2TiO_5$ powder had a rhombic-shaped morphology and different sized agglomerates consisting of smaller particles. This could be due to the coalescence of small particles into agglomerates [28,29]. Analysis of periodic lattice fringes in the high-resolution transmission electron microscopy (HRTEM) image shown in figure 2*c* was performed using a fast Fourier transform (FFT). The calculated FFT patterns enabled the determination of the lattice spacing

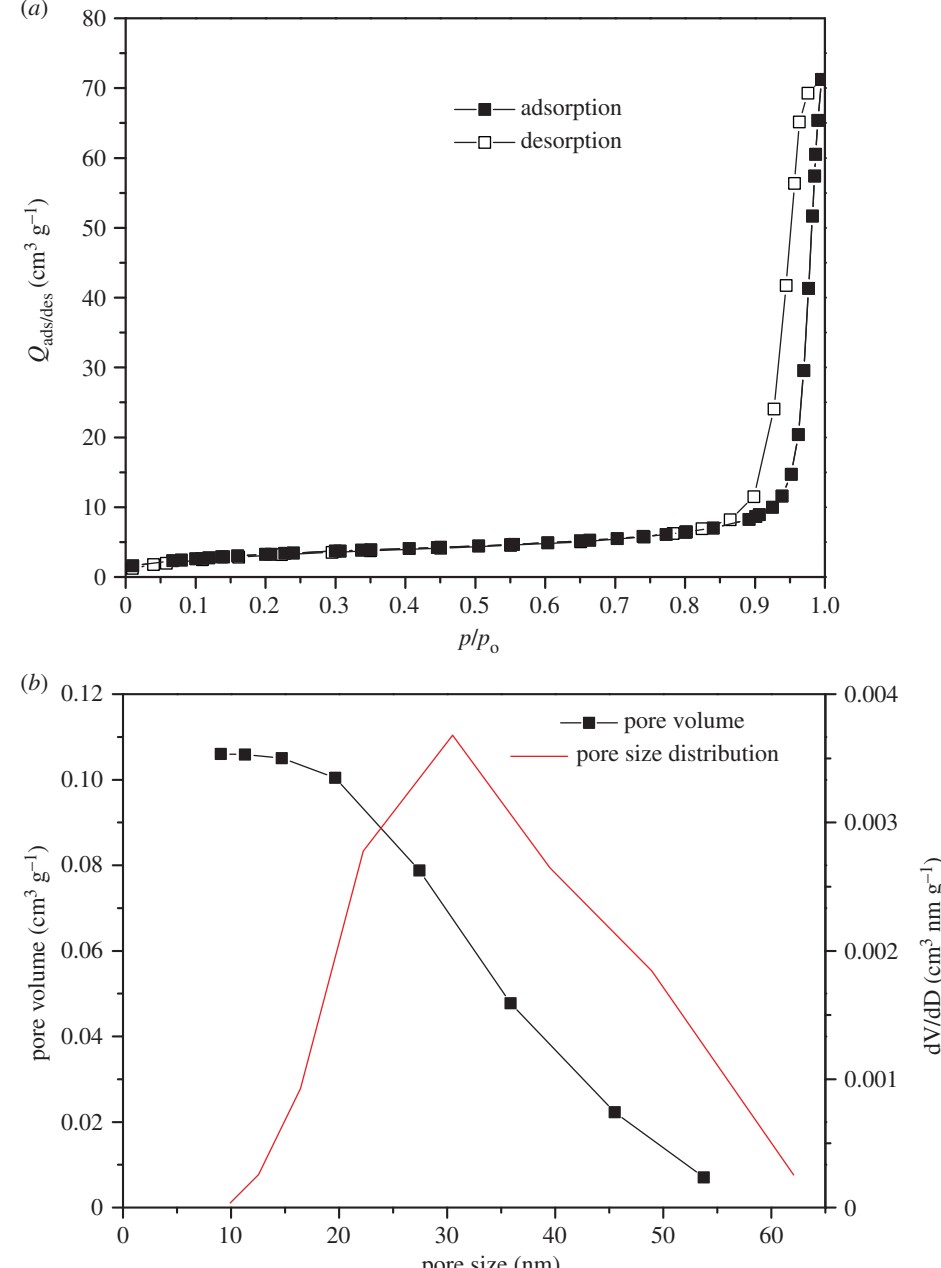

**Figure 3.** Nitrogen adsorption–desorption isotherms (*a*) and BET pore size distribution curves (*b*) for Fe$_2$TiO$_5$ powder.

**Table 2.** Textural properties for the Fe$_2$TiO$_5$ powder.[a]

| sample | $S_{BET}$ (m$^2$ g$^{-1}$) | $V_{tot}$ (cm$^3$ g$^{-1}$) | $V_{meso}$ (cm$^3$ g$^{-1}$) | $V_{micro}$ (cm$^3$ g$^{-1}$) | $D_{av}$ (nm) | $D_{max}$ (nm) |
|---|---|---|---|---|---|---|
| Fe$_2$TiO$_5$ | 11.8 | 0.1072 | 0.1060 | 0.0035 | 31.0 | 30.5 |

[a]Includes $S_{BET}$—BET specific surface area, $V_{tot}$—total volume of the pores, $V_{meso}$—volume of the mesopores $V_{micro}$—volume of the micropores, $D_{av}$—the mean pore diameter, $D_{max}$—diameter of the pores occupying the greatest portion of the volume.

of 0.49 nm (marked in figure 2*c*) that corresponds to the (020) plane of an orthorhombic *Cmcm* Fe$_2$TiO$_5$ structure in accordance with the parameters determined by Rietveld analysis of the measured X-ray diffraction pattern.

## 3.3. Texture

Figure 3*a* shows the N$_2$ adsorption–desorption isotherm of Fe$_2$TiO$_5$ powder. According to the IUPAC nomenclature [30], the shape of the nitrogen isotherm could be considered to be a type V isotherm,

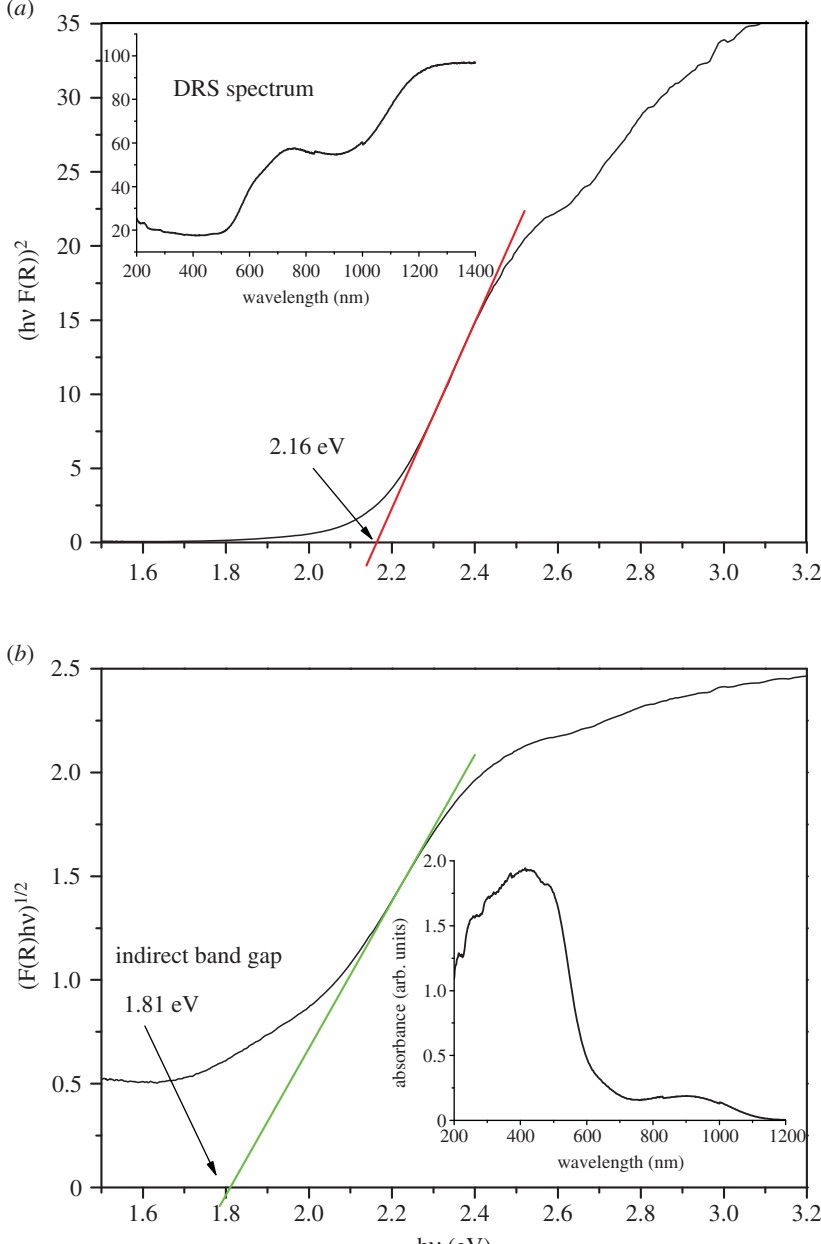

**Figure 4.** Tauc plot for estimation of direct (*a*) and indirect (*b*) band gaps of Fe$_2$TiO$_5$ powder; measured diffuse reflectance spectrum, inset (*a*), absorbance obtained by Kubelka–Munk approximation, inset (*b*).

which is typical for mesoporous materials. N$_2$ adsorption–desorption isotherm data for Fe$_2$TiO$_5$ powder are depicted in table 2. The specific surface area of the examined sample was 11.8 m$^2$ g$^{-1}$ and the total pore volume was found to be 0.1072 cm$^3$ g$^{-1}$. The presence of mesopores [31] with pore diameters ranging from 10 to 60 nm and D$_{max}$ approximately 30.5 nm (figure 3*b*) was confirmed from pore size distribution analysis.

## 3.4. Band gap

The measured diffraction reflection spectrum (DRS) of Fe$_2$TiO$_5$ powder is shown as an inset in figure 4*a*. It can be seen that Fe$_2$TiO$_5$ nanoparticles have strong visible-light absorption. The equivalent absorption coefficient was calculated using the Kubelka–Munk function and is shown as an inset in figure 4*b*. Tauc plots for a direct (figure 4*a*) and indirect (figure 4*b*) system [32] enabled estimation of the optical band gap values using the relationship of Davis and Mott [33]. Direct and indirect band gap values were estimated as 2.16 and 1.81 eV, respectively. These are similar to some previously obtained values for

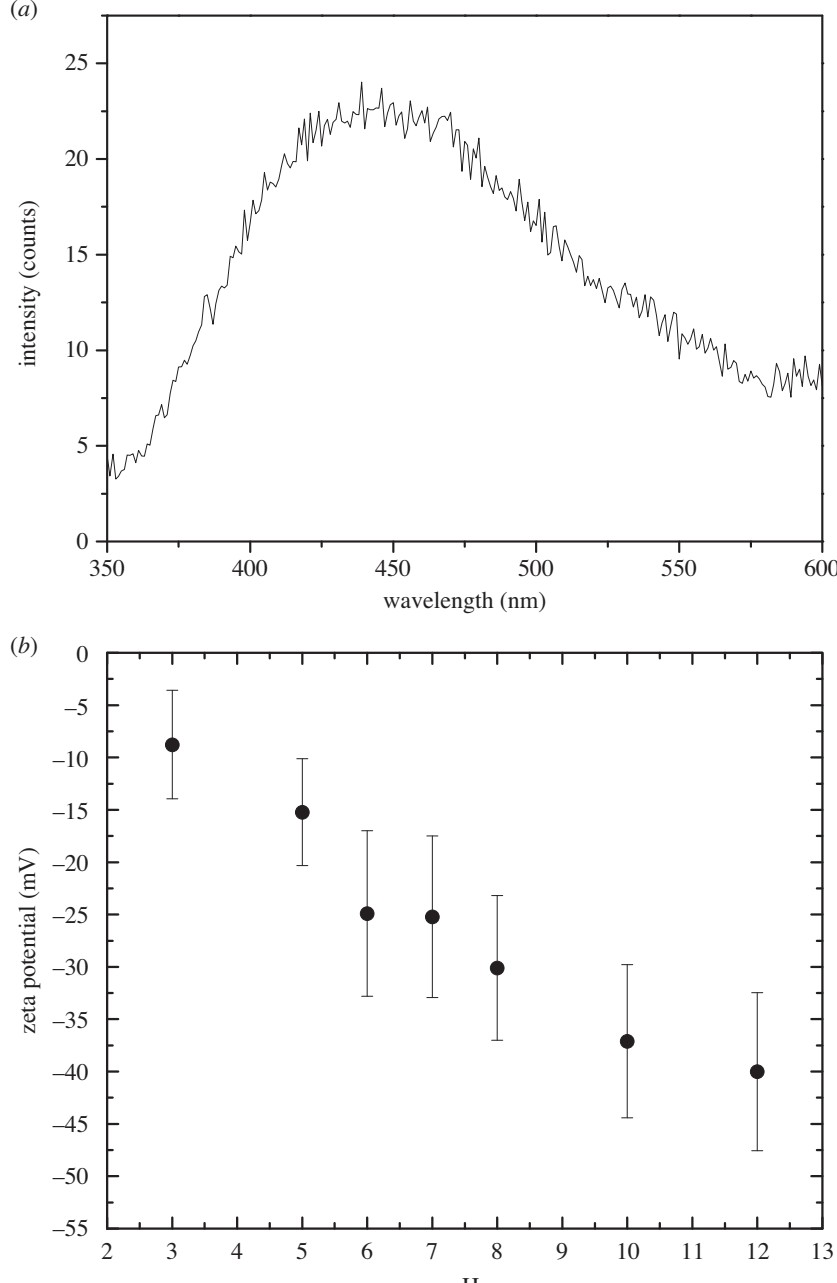

**Figure 5.** Photoluminescence spectrum measured with 270 nm excitation wavelength (*a*) and zeta potential (*b*) determined for $Fe_2TiO_5$ nanoparticles.

$Fe_2TiO_5$ [34,35], though many authors have experimentally and by DFT calculations obtained higher values for the indirect band gap in the range 2–2.2 eV [24,27,36–38]. Considering these results, as synthesized $Fe_2TiO_5$ nanoparticles could have the potential for the degradation of different pollutants in the visible-light region.

## 3.5. Photoluminescence and colloidal properties

Photoluminescence spectrum of $Fe_2TiO_5$ nanoparticle powder measured with an excitation wavelength of 270 nm is shown in figure 5*a*. Overall low values were obtained indicating that recombination of the photo-generated electron–hole pairs is inhibited. The separation efficiency of photo-excited electron–hole pairs is enhanced leading to improved photocatalytic properties [39]. This has been observed before for $Fe_2TiO_5/TiO_2$ nanoheterostructures and composites [39,40].

Determining the point of zero charge ($pH_{PZC}$) is significant to predict the charge on the nanoparticle surface during photocatalytic degradation of pollutants [41]. The stability of as-synthesized $Fe_2TiO_5$ particles was analysed by measuring the zeta potential in an aqueous system using a deionized water suspension at room temperature. When metal oxide nanoparticles are dispersed in water or an organic solvent, –OH surface groups can be protonated or deprotonated leading to surface charged groups ($-OH_2^+$ or $-O^-$), respectively. The zeta potential observed for $Fe_2TiO_5$ particles was $-21.6 \pm 6.8$ mV. This value is negative and high and indicates particle stability in water and low agglomeration [42]. Adsorption of cationic dyes such as MB on a negative surface charge is favoured due to electrostatic forces of attraction (Van der Waals) [43]. Variation of the zeta potential with pH is given in figure 5b. All values were negative showing that the point of zero charge ($P_{ZC}$) or isoelectric point (IEP) of pseudobrookite is below pH 3. The zeta potential value can be an indicator of the particle surface charge and is influenced by the preparation process [44]. Muthukumar *et al.* [44] obtained high positive values for pseudobrookite (iron titanate) synthesized using a solvothermal method that depended on the duration of treatment in reduced oxygen conditions (19.4 for 5 h, 19.98 for 10 h, 33.6 for 15 h and 17.43 for 20 h). Such values enabled acidic site availability. In our case, the obtained gel was calcined in an air atmosphere rich in oxygen that resulted in a negative charge. Mehdilo *et al.* [45] measured the zeta potential of ilmenite ore (raw, heat-treated (fired) at 500, 600, 750 and 950°C). Compared with raw ilmenite with a point of zero charge ($P_{ZC}$) determined at pH 5.4, heat treating of ilmenite at 600°C accompanied with the decomposition of ilmenite to haematite and rutile (at 750°C) and then ferric pseudobrookite ($Fe_2TiO_5$), haematite and rutile at 950°C lead to an overall decrease in zeta potential values over the pH range from 2.5 to 7.5 and shift of the $P_{ZC}$ to pH 3.9, 2.9 and 3 at 600, 750 and 950°C, respectively. Suttiponparnit *et al.* [46] investigated the role of surface area, primary particle size and crystal phase on $TiO_2$ dispersion properties and determined that in the case of anatase, increase in particle size leads to a reduction of the dispersion zeta potential and increase in hydrodynamic size, with a completely negative zeta potential for particles of 102 nm. This could also be the case for $Fe_2TiO_5$ and needs further investigation on different-sized particles.

The hydrodynamic radius of $Fe_2TiO_5$ particles was determined as $175 \pm 7$ nm. Taking into account that the average particle size was estimated to be around 50 nm in FESEM images and that agglomerates were noted in TEM images, this value shows that some particle aggregation remained.

## 3.6. Photodegradation of methylene blue

The photocatalytic activity of $Fe_2TiO_5$ was evaluated by the degradation of MB aqueous solution under direct sunlight.

### 3.6.1. Effect of initial photocatalyst concentration on methylene blue degradation

The effect of initial photocatalyst concentration under direct sunlight irradiation was investigated by varying the initial $Fe_2TiO_5$ concentration from 10 to 50 mg per 100 ml of 10 mg l$^{-1}$ MB dye solution as shown in figure 6a. It can be seen from figure 6b that the dye removal efficiency without $Fe_2TiO_5$ was negligible which is consistent with previously reported literature data [47]. When the amount of photocatalyst was increased from 10 to 50 mg, the photocatalytic activity was found to be increasing, which could be caused by an increased number of active sites on the catalyst surface. Further increase in the catalyst amount above 50 mg decreased the activity, which could be due to the phenomenon of light scattering and screening effects (electronic supplementary material, S.1). Namely, this can be ascribed to the increased aggregation of particles acting as barriers for the light irradiation [48,49].

### 3.6.2. Effect of the initial dye concentration

The effect of the initial concentration of MB on the decomposition of the dye was investigated by varying the initial concentration from 10 to 30 mg l$^{-1}$ with constant catalyst loading (50 mg). As shown in electronic supplementary material S.2, the photodegradation process decreased with the increase in dye concentration. This can be explained as follows: due to increased adsorption of dye molecules on surface-active sites on the photocatalyst, adsorption of $OH^-$ is decreased, and therefore the formation of the highly oxidative, $OH\cdot$ radical is reduced. In addition, higher dye concentrations reduce the light path length of the photons thus the light-triggered catalyst decreases [50,51].

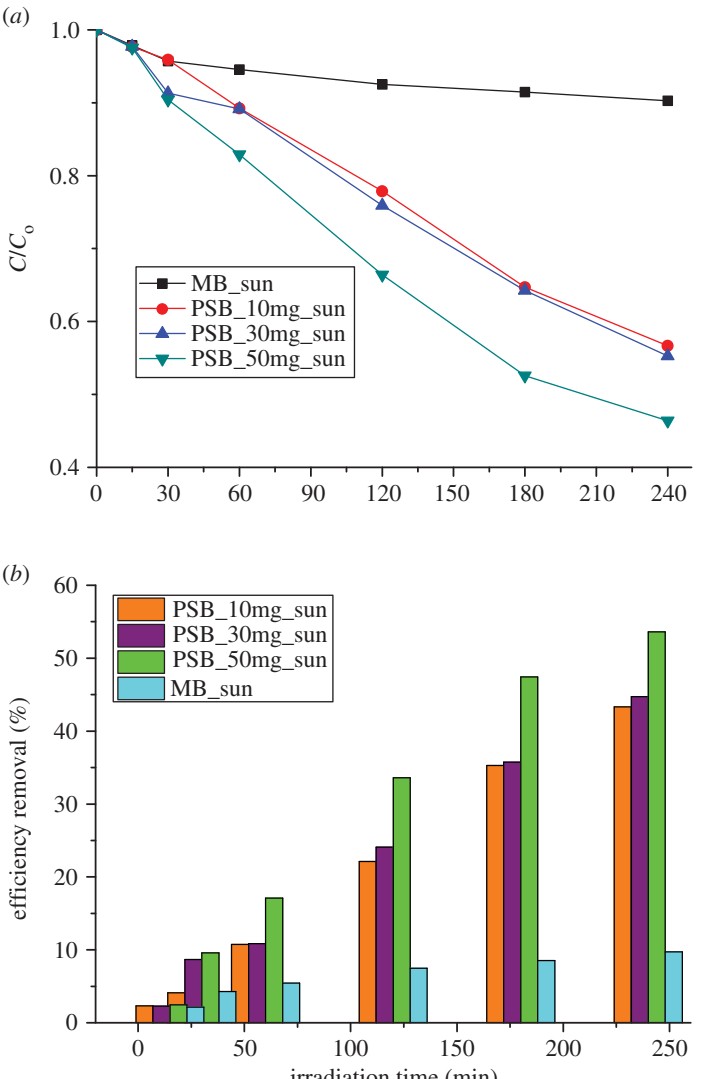

**Figure 6.** Effect of initial photocatalyst dose on the degradation of MB (*a*) and percentage removal of MB (*b*).

### 3.6.3. Effect of solution pH on the degradation of methylene blue

The pH is considered to be the main parameter in the photocatalytic process, due to its impact on the surface charge of the photocatalyst affecting dye degradation [52]. Hence, an attempt was made to study the influence of pH on the degradation of MB at pH values in the range of 7–11 (figure 7*a*), and it was adjusted before adsorption. We also investigated photocatalytic degradation in acidic conditions (pH = 3), where the influence was negligible. The degradation efficiency increased with increased pH (97% at pH = 11) which is 1.79 times higher than the degradation efficiency at the normal pH condition. The corresponding colour change is given in electronic supplementary material, S.3. Since the zeta potential was negative with pH change, it is expected that adsorption of MB (cationic dye) on the surface of the $Fe_2TiO_5$ photocatalyst would be enhanced at higher pH values [1,53]. Namely, in the photocatalytic oxidation process after irradiation and excitation of electrons from the valence to the conduction band (figure 8), photo-generated holes either directly oxidize dye to reactive intermediates (equation (3.2)) or react with hydroxyl ions ($OH^-$) leading to the formation of highly oxidative hydroxyl radicals ($OH\cdot$) and further complete mineralization of the dye (equation (3.5)) [54].

$$Fe_2TiO_5 + hv \rightarrow Fe_2TiO_5(e_{CB}^- + h_{VB}^+) \tag{3.1}$$

and

$$h_{VB}^+ + MB \rightarrow MB^{\cdot+} \rightarrow oxidation\ of\ MB, \tag{3.2}$$

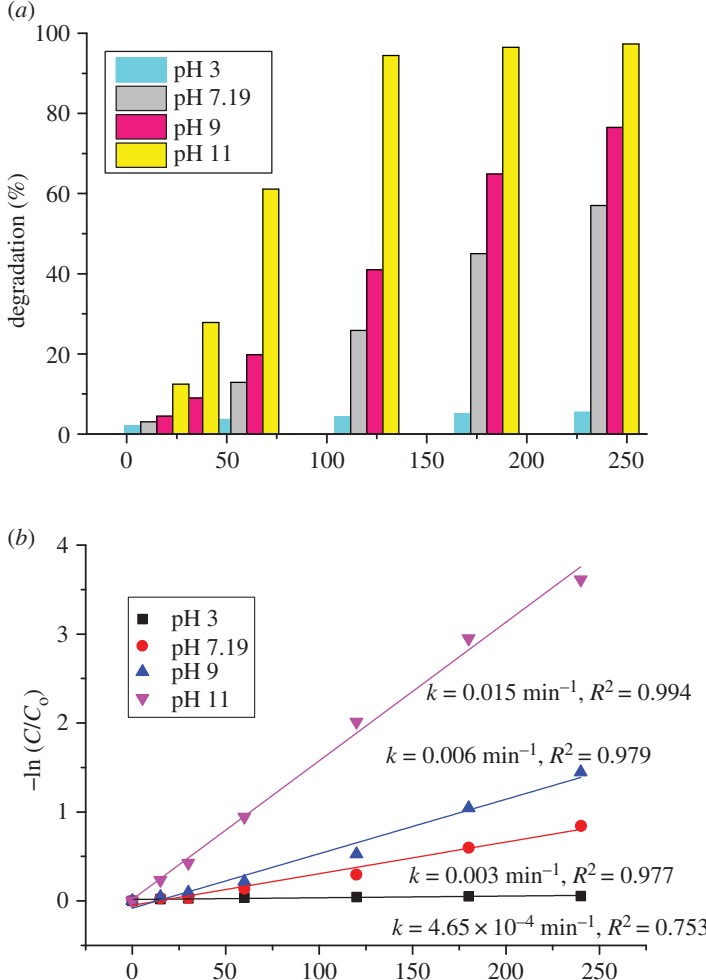

**Figure 7.** Efficiency of removal of MB—effect of the initial pH on the degradation of MB (a) and pseudo-first-order kinetics curves of the degradation of MB under sunlight in the presence of $Fe_2TiO_5$ with different initial pH values (b). Experimental conditions: $Fe_2TiO_5$ 50 mg $l^{-1}$ and MB 10 mg $l^{-1}$.

or

$$h_{VB}^+ + H_2O(OH^-) \rightarrow OH^. + H^+ \tag{3.3}$$

and

$$OH^. + MB \; dye \rightarrow CO_2 + H_2O. \tag{3.4}$$

However, since the adsorption of dye was low as the surface area was moderate (11.8 $m^2 \, g^{-1}$), a possible reason for enhanced photocatalytic degradation of MB at higher pH values is the formation of OH radicals due to a large number of hydroxyl ions in the alkaline medium [55].

A simplified Langmuir–Hinshelwood (L–H) kinetic model (equation 3.5) was used to describe the photocatalytic degradation rate of MB by plotting the graph of $-\ln(C/C_0)$ versus time, $t$, (figure 7b) at different pH values [49].

$$-\ln\frac{C}{C_o} = kt, \tag{3.5}$$

where $C_0$ and C are the concentrations of MB in solution at time 0 and $t$, respectively, and $k$ is the apparent first-order reaction rate constant. The obtained values of $k$ are equal to $4.65 \times 10^{-4}$, 0.00356, 0.00612 and 0.016 $min^{-1}$ for initial pH values 3, 7.19, 9 and 11, while correlation constants, $R^2$, for the fitted lines were 0.753, 0.977, 0.979 and 0.994, respectively.

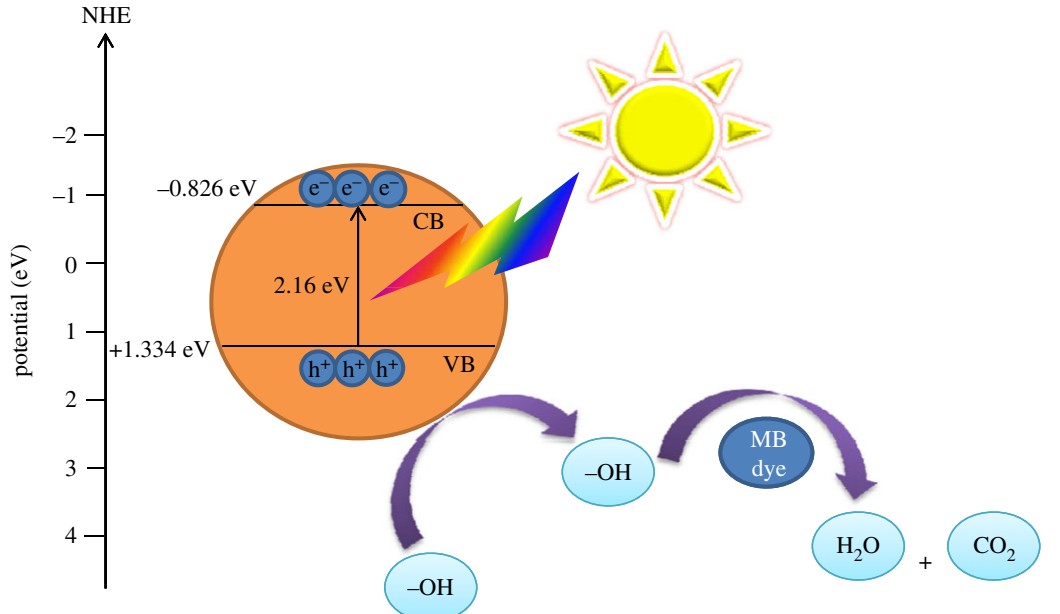

**Figure 8.** Scheme of the band gap structure of $Fe_2TiO_5$ and the possible mechanism for the photocatalytic degradation of MB dye over $Fe_2TiO_5$ photocatalyst.

### 3.6.4. Possible mechanism of methylene blue degradation

Efficient charge separation is the key factor determining the photocatalytic activity of a semiconducting photocatalyst. Thus, it was essential to determine the conduction band (CB) and valence band (VB) potentials of the $Fe_2TiO_5$ photocatalyst, using the following empirical equations [56]:

$$E_{VB} = \chi + 0.5E_g - E^e \tag{3.6}$$

and

$$E_{CB} = E_{VB} - E_g, \tag{3.7}$$

where $E_{VB}$ is the valence band (VB) edge potential, $E_{CB}$ is the conduction band (CB) edge potential, $\chi$ is the electronegativity of the semiconductor, $E^e$ is the energy of free electrons on the hydrogen scale (4.5 eV) and $E_g$ is the band gap energy of the semiconductor. Electronegativity of $Fe_2TiO_5$ was calculated using the following equation:

$$\chi = [\chi(A)^a \chi(B)^b \chi(C)^c]^{1/a+b+c}, \tag{3.8}$$

where $a$, $b$ and $c$ are the number of atoms in the compound, while electronegativity of Fe, Ti and O was calculated as the arithmetic mean of the atomic electron affinity and the first ionization energy [57]. The value of $\chi$ of $Fe_2TiO_5$ was calculated to be 2.086 while $E_{CB}$ and $E_{VB}$ were estimated to be −0.826 and + 1.334 eV versus normal hydrogen electrode (NHE) and results are presented in figure 8. The conduction band of $Fe_2TiO_5$ is more negative than that of $O_2/\cdot O_2^-$ (−0.282 eV versus NHE); therefore, the electrons can reduce $E_{VB}$ $O_2$ to generate superoxide anions $\cdot O_2^-$, while the $E_{VB}$ of $Fe_2TiO_5$ (+1.334 eV versus NHE) was not positive enough to oxidize $OH^-/H_2O$ (2.27 V versus NHE) to form active species $\cdot OH$ [58]. This indicates that MB dye was degraded mainly by the active species $\cdot O_2^-$ and by the accumulation of holes. Enhanced photocatalytic degradation efficiency could be ascribed to a large number of hydroxyl ions in the alkaline medium, as mentioned above.

## 4. Conclusion

In summary, $Fe_2TiO_5$ nanopowder was successfully synthesized through a simple sol–gel route followed by calcination at 750°C. The $Fe_2TiO_5$ powder consisted of relatively uniformly sized nanoparticles with a rhombic shape and an average diameter of around 50 nm. The specific surface area from BET analysis was around 11.8 $m^2\,g^{-1}$. The band gap was estimated to be 2.16 eV. The zeta potential observed for $Fe_2TiO_5$ was negative and high and indicated particle stability in water and low agglomeration.

Photocatalytic measurements showed that as-synthesized $Fe_2TiO_5$ displays moderate degradation efficiency. The degradation of efficiency significantly depends on the particle specific surface area. In addition, catalyst loading and pH of the solution greatly affected the degradation process. A reaction mechanism was proposed.

Data accessibility. Material characterization results and methylene blue aqueous solution degradation data are available at the Dryad Digital Repository (https://doi.org/10.5061/dryad.69p8cz8z5) [59]. Additional information concerning this paper is available in the electronic supplementary material

Authors' contributions. Z.Z.V. conceived and designed the study, synthesized the powder, performed photocatalytic experiments and spectroscopic analysis, and drafted the manuscript. M.P.D. participated in the laboratory work, especially photocatalytic experiments and spectroscopic analysis, and critically revised the manuscript. J.D.V. measured and analysed FESEM images. I.J.-C. performed measurement of nitrogen adsorption/desorption isotherms and analysed obtained data. M.O. measured and analysed the zeta potential and particle size. N.B.T. measured X-ray diffraction spectra. S.S. measured and analysed photoluminescence spectra. G.O.B. measured and analysed TEM images. M.V.N. helped conceive and design the study, coordinated the study, analysed XRD data and TEM images, measured and analysed FTIR spectra and helped draft the manuscript. All authors gave final approval for publication and agree to be held accountable for the work performed therein.

Competing interests. We declare we have no competing interests.

Funding. This work was funded by the Ministry for Education, Science and Technological Development of the Republic of Serbia with contracts for realization and financing scientific research with the Institute of Technical Sciences of SASA no. 451-03-68/2020-14/200175 (Z.Z.V. and J.D.V.) and with the Institute for Multidisciplinary Research, University of Belgrade, no. 451-03-68/2020-14/200053 (M.P.D., G.O.B. and M.V.N.)

Acknowledgements. The support of the bilateral cooperation with Slovenia is also gratefully acknowledged (project no. BI-RS-18-19-026). The TEM work was conducted in the Centre for Electron Microscopy and Microanalysis (CEMM) at Jožef Stefan Institute, Ljubljana (Slovenia).

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
