## [Reviewer comments · Royal Society Open Science]

Review History

RSOS-200708.R0 (Original submission)

Review form: Reviewer 1

Is the manuscript scientifically sound in its present form?

Yes

Are the interpretations and conclusions justified by the results?

Yes

Is the language acceptable?

No

Do you have any ethical concerns with this paper?

No

Have you any concerns about statistical analyses in this paper?

Yes

Recommendation?

Major revision is needed (please make suggestions in comments)

Comments to the Author(s)

In this work, the authors reported on the synthesis of Iron (Fe_2TiO_5) using a sol-gel and calcination at 750°C . The synthesized nano particles were investigated by necessary tools where size and morphology have been represented. the work is worth publishing however, some comments and corrections need to be addressed by the authors before acceptance.

- English language needs polishing particularly the introduction section.

- please add the aim of the work to the abstract.

- please use a subscript for chemical formula ($\text{Fe}(\text{NO})_9\text{H}_2\text{O}$)

- please give the resolution of the SEM & TEM images.

- please provide statistical analysis

i recommend major revision.

Review form: Reviewer 2**Is the manuscript scientifically sound in its present form?**

No

Are the interpretations and conclusions justified by the results?

No

Is the language acceptable?

Yes

Do you have any ethical concerns with this paper?

No

Have you any concerns about statistical analyses in this paper?

No

Recommendation?

Major revision is needed (please make suggestions in comments)

Comments to the Author(s)

In this present article, authors have used sol gel method to synthesized the Fe_2TiO_5 nanoparticles, which were calcined at high temperature later. Various physical and chemical characterization techniques are reported in order to evaluate the nanoparticles for its practical use to as a photocatalyst to fragment the dye molecule (MB) in natural light. The research paper can be accepted once following issues are completely addressed.

1. Page No 3 of 12 line 60 please "Zeta potential and particle size by dynamic light scattering (DLS) at" confirm the temperature value. Like that there are many mistakes (for e.g. Page No. 4 of 12 line 8 and 9) in the entire draft which has to be corrected before acceptance of publications.

2. Figure 1 (a) X-ray diffraction pattern of synthesized Fe_2TiO_5 nanocrystalline powder, inset: Rietveld analysis....

Authors are suggested to use Rietveld analysis in the same figure instead of showing it separately as inset. They are suggested to index the remaining XRD peaks as observed. Further in analysis part authors confirms the single phase Fe_2TiO_5 nanostructures, where the grain size reported herewith is estimated using the Scherrer method. Here authors are suggested to use Williamson

and Hall (W-H) method to estimate the grain size and the crystal distortions if any to report these values more accurately.

3. Page 5 of 15, line 34 ...figure 3a reveals the FESEMhowever in actual text it is "Figure 2a", authors suggested to correct it accordingly. Authors are suggested to present high quality HRTEM images where the fringes can be clearly observed.
4. Rom Fig. 6 (a) it reflects that the blank text to evaluate photolysis indicate increase in the absorbance with the expose time ...it is recommended to provide discussion for the same.
5. The authors should provide a scheme regarding the position of the bands for the composite synthesized to explain the photocatalytic activity as claimed.
6. Authors are expected to comment on the effects of self-sensitisation, control photocatalytic reactions run using the colourless organic model pollutants like phenol, 4-chlorophenol or catechol.

Decision letter (RSOS-200708.R0)

Dear Dr Vasiljevic:

Title: Photocatalytic degradation of methylene blue under natural sunlight using Fe₂TiO₅ nanoparticles prepared by a modified sol-gel method
Manuscript ID: RSOS-200708

The editor assigned to your manuscript has now received comments from reviewers. We would like you to revise your paper in accordance with the referee and Subject Editor suggestions which can be found below (not including confidential reports to the Editor). Please note this decision does not guarantee eventual acceptance.

Please submit your revised paper before 18-Jul-2020. Please note that the revision deadline will expire at 00.00am on this date. If we do not hear from you within this time then it will be assumed that the paper has been withdrawn. In exceptional circumstances, extensions may be possible if agreed with the Editorial Office in advance. We do not allow multiple rounds of revision so we urge you to make every effort to fully address all of the comments at this stage. If deemed necessary by the Editors, your manuscript will be sent back to one or more of the original reviewers for assessment. If the original reviewers are not available we may invite new reviewers.

On behalf of the Subject Editor Professor Anthony Stace and the Associate Editor Dr Dattatray Late.

RSC Associate Editor:

Comments to the Author:

Fe₂TiO₅ nanopowder synthesis and its application for dye degradation were reported. Authors need to revise the manuscript as per Reviewer's suggestions.

RSC Subject Editor:

Comments to the Author:

Reviewers' Comments to Author:

Reviewer: 1

Comments to the Author(s)

In this work, the authors reported on the synthesis of Iron (Fe₂TiO₅) using a sol-gel and calcination at 750 °C. The synthesized nano particles were investigated by necessary tools where size and morphology have been represented. The work is worth publishing however, some comments and corrections need to be addressed by the authors before acceptance.

- English language needs polishing particularly the introduction section.

- please add the aim of the work to the abstract.

- please use a subscript for chemical formula (Fe(NO)₃ · 9H₂O)

- please give the resolution of the SEM & TEM images.

- please provide statistical analysis

I recommend major revision.

Reviewer: 2

Comments to the Author(s)

In this present article, authors have used sol gel method to synthesize the Fe₂TiO₅ nanoparticles, which were calcined at high temperature later. Various physical and chemical characterization techniques are reported in order to evaluate the nanoparticles for its practical use as a photocatalyst to fragment the dye molecule (MB) in natural light. The research paper can be accepted once following issues are completely addressed.

1. Page No 3 of 12 line 60 please “Zeta potential and particle size by dynamic light scattering (DLS) at” confirm the temperature value. Like that there are many mistakes (for e.g. Page No. 4 of 12 line 8 and 9) in the entire draft which has to be corrected before acceptance of publications.
2. Figure 1 (a) X-ray diffraction pattern of synthesized Fe₂TiO₅ nanocrystalline powder, inset: Rietveld analysis....
Authors are suggested to use Rietveld analysis in the same figure instead of showing it separately as inset. They are suggested to index the remaining XRD peaks as observed. Further in analysis part authors confirms the single phase Fe₂TiO₅ nanostructures, where the grain size reported herewith is estimated using the Scherrer method. Here authors are suggested to use Williamson and Hall (W-H) method to estimate the grain size and the crystal distortions if any to report these values more accurately.
3. Page 5 of 15, line 34 ...figure 3a reveals the FESEMhowever in actual text it is “Figure 2a”, authors suggested to correct it accordingly. Authors are suggested to present high quality HRTEM images where the fringes can be clearly observed.
4. Rom Fig. 6 (a) it reflects that the blank text to evaluate photolysis indicate increase in the absorbance with the expose time ...it is recommended to provide discussion for the same.
5. The authors should provide a scheme regarding the position of the bands for the composite synthesized to explain the photocatalytic activity as claimed.
6. Authors are expected to comment on the effects of self-sensitisation, control photocatalytic reactions run using the colourless organic model pollutants like phenol, 4-chlorophenol or catechol.

Author's Response to Decision Letter for (RSOS-200708.R0)

See Appendix A.

RSOS-200708.R1 (Revision)

Review form: Reviewer 1

Is the manuscript scientifically sound in its present form?

Yes

Are the interpretations and conclusions justified by the results?

Yes

Is the language acceptable?

Yes

Do you have any ethical concerns with this paper?

No

Have you any concerns about statistical analyses in this paper?

No

Recommendation?

Accept as is

Comments to the Author(s)

i recommend acceptance of the revised form.

Review form: Reviewer 2**Is the manuscript scientifically sound in its present form?**

Yes

Are the interpretations and conclusions justified by the results?

Yes

Is the language acceptable?

Yes

Do you have any ethical concerns with this paper?

No

Have you any concerns about statistical analyses in this paper?

Yes

Recommendation?

Accept as is

Comments to the Author(s)

All queries are addressed, hence accept the paper for possible publication as it is

Decision letter (RSOS-200708.R1)

Dear Dr Vasiljevic:

Title: Photocatalytic degradation of methylene blue under natural sunlight using Fe₂TiO₅ nanoparticles prepared by a modified sol-gel method
Manuscript ID: RSOS-200708.R1

It is a pleasure to accept your manuscript in its current form for publication in Royal Society Open Science. The chemistry content of Royal Society Open Science is published in collaboration with the Royal Society of Chemistry.

Yours sincerely,
Dr Laura Smith

Publishing Editor, Journals

On behalf of the Subject Editor Professor Anthony Stace and the Associate Editor Dr Dattatray Late.

RSC Associate Editor:
Comments to the Author:
(There are no comments.)

RSC Subject Editor:
Comments to the Author:
(There are no comments.)

Reviewer(s)' Comments to Author:
Reviewer: 1

Comments to the Author(s)
i recommend acceptance of the revised form.

Reviewer: 2

Comments to the Author(s)
All queries are addressed, hence accept the paper for possible publication as it is

Appendix A

We would like to thank the reviewers for their useful comments that have helped us improve the quality of our paper.

Reviewers' Comments to Author:

Reviewer: 1

Comments to the Author(s)

In this work, the authors reported on the synthesis of Iron (Fe_2TiO_5) using a sol-gel and calcination at 750 °C. The synthesized nano particles were investigated by necessary tools where size and morphology have been represented. The work is worth publishing however, some comments and corrections need to be addressed by the authors before acceptance.

- English language needs polishing particularly the introduction section.

Thank you for your comments. We have revised and polished the paper, especially the Introduction, all changes are marked in red.

- please add the aim of the work to the abstract.

Thank you for your comments. We have added the aim of the work to the manuscript.

- please use a subscript for chemical formula ($\text{Fe}(\text{NO})_3 \cdot 9\text{H}_2\text{O}$)

Thank you for your comments. We have added the subscript for the chemical formula.

- please give the resolution of the SEM & TEM images.

Thank you for your comments. We have added the resolution of the SEM and TEM images to the figure caption.

- please provide statistical analysis

Thank you for your comments. We have counted 72 particles from 6 TEM images and calculated the mean size as 49.7 nm with a standard deviation of 12.6 nm, and have added this description to the text.

i recommend major revision.

Reviewer: 2

Comments to the Author(s)

In this present article, authors have used sol gel method to synthesized the Fe_2TiO_5 nanoparticles, which were calcined at high temperature later. Various physical and chemical characterization techniques are reported in order to evaluate the nanoparticles for its practical use to as a photocatalyst to fragment the dye molecule (MB) in natural light. The research paper can be accepted once following issues are completely addressed.

1. Page No 3 of 12 line 60 please "Zeta potential and particle size by dynamic light scattering (DLS) at" confirm the temperature value. Like that there are many mistakes (for e.g. Page No. 4 of 12 line 8 and 9) in the entire draft which has to be corrected before acceptance of publications.

Thank you for your comments. We have corrected all the mistakes in the text concerning symbols, including the ones noted by the reviewer .

2. Figure 1 (a) X-ray diffraction pattern of synthesized Fe_2TiO_5 nanocrystalline powder, inset: Rietveld analysis...

Authors are suggested to use Rietveld analysis in the same figure instead of showing it separately as inset. They are suggested to index the remaining XRD peaks as observed. Further in analysis part authors confirms the single phase Fe_2TiO_5 nanostructures, where the grain size reported herewith is estimated using the Scherrer method. Here authors are suggested to use Williamson and Hall (W-H) method to estimate the grain size and the crystal distortions if any to report these values more accurately.

Thank you for the comments. We have calculated the crystallite size using the Williamson and Hall method as suggested by the reviewer and obtained the crystallite size of 48.77 nm and have added this to

the text. We have also changed-redrawn Figure 1a to place Rietveld analysis in the same figure (not inset) and have indexed all the remaining peaks.

3. Page 5 of 15, line 34 ...figure 3a reveals the FESEMhowever in actual text it is "Figure 2a", authors suggested to correct it accordingly. Authors are suggested to present high quality HRTEM images where the fringes can be clearly observed.

Thank you for your comments. We have corrected the figure number in the text. In this work we have given the best HRTEM images that were recorded in Slovenia and we currently have no possibility to repeat these measurements.

4. Rom Fig. 6 (a) it reflects that the blank text to evaluate photolysis indicate increase in the absorbance with the expose time ...it is recommended to provide discussion for the same.

Thank you for your comments. In our case, during photolysis under sun irradiation, absorbance decreased. Experimental data have been uploaded in Figure 6a

(https://datadryad.org/stash/share/NVAh_fH9wg45B5VWQl_N-fUs9XErVxOrB1ITMpKUSqs)

5. The authors should provide a scheme regarding the position of the bands for the composite synthesized to explain the photocatalytic activity as claimed.

Thank you for your comments. We have calculated the position of the bands using an empirical model described in the text and added the references we used. We have also drawn the scheme showing the position of the bands, and added it as figure 8 in the text.

6. Authors are expected to comment on the effects of self-sensitisation, control photocatalytic reactions run using the colourless organic model pollutants like phenol, 4-chlorophenol or catechol.

Thank you for your comments. Aim of our work was to investigate photocatalytic degradation of methylene blue as a model pollutant using Fe_2TiO_5 as photocatalyst synthesized via simple sol-gel route without any modification. Future work will be expanded on photocatalytic degradation of other organic pollutants, but also on optimization of synthesis of Fe_2TiO_5 as promising photocatalyst.